# Analysis of Human Papilloma Virus Content and Integration in Mucoepidermoid Carcinoma

**DOI:** 10.3390/v14112353

**Published:** 2022-10-26

**Authors:** Wenjin Gu, Apurva Bhangale, Molly E. Heft Neal, Josh D. Smith, Collin Brummel, Jonathan B. McHugh, Matthew E. Spector, Ryan E. Mills, J. Chad Brenner

**Affiliations:** 1Department of Bioinformatics and Computational Biology, University of Michigan, Ann Arbor, MI 48109, USA; 2Department of Otolaryngology–Head and Neck Surgery, University of Michigan, Ann Arbor, MI 48109, USA; 3Department of Pathology, University of Michigan, Ann Arbor, MI 48109, USA; 4Rogel Cancer Center, University of Michigan, Ann Arbor, MI 48109, USA; 5Department of Pharmacology, University of Michigan, Ann Arbor, MI 48109, USA

**Keywords:** MEC, HPV, CRTC1/3-MAML2, PIK3AP1, SIRT1

## Abstract

Mucoepidermoid Carcinomas (MEC) represent the most common malignancies of salivary glands. Approximately 50% of all MEC cases are known to harbor *CRTC1/3-MAML2* gene fusions, but the additional molecular drivers remain largely uncharacterized. Here, we sought to resolve controversy around the role of human papillomavirus (HPV) as a potential driver of mucoepidermoid carcinoma. Bioinformatics analysis was performed on 48 MEC transcriptomes. Subsequent targeted capture DNA sequencing was used to annotate HPV content and integration status in the host genome. HPV of any type was only identified in 1/48 (2%) of the MEC transcriptomes analyzed. Importantly, the one HPV16+ tumor expressed high levels of p16, had high expression of HPV16 oncogenes E6 and E7, and displayed a complex integration pattern that included breakpoints into 13 host genes including *PIK3AP1*, *HIPI,* *OLFM4,*
*SIRT1*, *ARAP2*, *TMEM161B-AS1,* and *EPS15L1* as well as 9 non-genic regions. In this cohort, HPV is a rare driver of MEC but may have a substantial etiologic role in cases that harbor the virus. Genetic mechanisms of host genome integration are similar to those observed in other head and neck cancers.

## 1. Introduction

Mucoepidermoid carcinomas (MEC) are the most common malignancies of the salivary glands comprising between 30–40% of all salivary gland cancers [1,2,3]. While MECs commonly arise in the parotid gland, they can occasionally form in other head and neck sites including the submandibular and sublingual glands, as well as the minor salivary glands of the oropharynx, oral cavity, and sinonasal cavities. Disease-specific survival is variable for patients with MEC and is dependent on factors such as histologic grade, tumor location, tumor stage, nodal status, patient age, margin status, and perineural invasion [2,3,4]. Importantly, however, it remains extremely challenging to differentiate between aggressive and non-aggressive MEC, which in the future may be improved by better understanding the molecular composition of this disease. In fact, a series of highly recurrent genetic alterations in MEC that lead to a Chr(11;19) (q14–21; p12–13) rearrangement and induce the formation of a *CRTC1*-*MAML2* fusion gene is one of the most widely studied alterations in this disease [5,6,7,8,9]. At present however, the prognostic significance of the *CRTC1-MAML2* gene fusion in MEC is unclear, further supporting the need to better define molecular drivers of the disease [6,9].

Given both the anatomic distribution of MEC primary sites and the well-established role of high-risk human papillomavirus (HPV) as drivers of certain head and neck squamous cell carcinoma (HNSCC), it is interesting to speculate whether HPV may be associated with MEC as another potential molecular driver. Accordingly, given the multiple studies showing a strong prognostic role of HPV in HNSCC (e.g., [10,11]) as well as the emerging role of HPV ctDNA in HNSCC disease monitoring [12,13,14], it is important to evaluate whether HPV content could have a similar molecular role in MEC. Unfortunately, however, there has been controversy in the literature about the role of high-risk HPV in MEC. Indeed, a 2011 study by Brunner et al. demonstrated that 2/6 (33.3%) of MEC cases analyzed contained high-risk HPV DNA by in situ hybridization as well as diffuse p16 overexpression [15]. In contrast, a study by Isayeva et al. used nested RT-PCR on RNA extracted from 98 MEC samples to show a much higher HPV positivity rate of 35/98 (36%) (where 23% of tumors contained HPV16, 6% contained HPV18, and 7% contained both HPV16 and HPV18) [16]. These authors presented orthogonal data using several HPV detection approaches including in situ hybridization, HPV16/18 E6 immunohistochemistry, and additional PCR-based validations to support the presence of HPV in their cohort. In strong contrast to this data, Bishop et al. used RNAscope-based ISH analysis with the HPV HR [17] probe set to demonstrate that a cohort of 71 MEC cases were all HPV negative [17]. The authors concluded that HPV does not appear to have any etiologic role in MEC carcinogenesis, independent of *CRTC1-MAML2* fusion status, though they acknowledged that their data conflicted with Isayeva et al. 

Given discrepant data around the prevalence of HPV in MEC, we sought to leverage our recently published MEC transcriptome data and advanced bioinformatics techniques to clarify the prevalence and status of HPV in MEC.

## 2. Materials and Methods

### 2.1. Clinical Specimens and Annotation of Viral Genomes 

A retrospective cohort of patients with MEC was previously identified from the University of Michigan pathology archive using an Institutional Review Board (IRB)-approved protocol for next generation sequencing of DNA and RNA (HUM00080561). However, patients were not consented for deposit of data in public databases. The cohort was previously typed for *CRTC1/3-MAML2* gene fusion status by RT-qPCR [18]. As previously noted, clinical, histologic, and outcomes data were collected from medical records and death was documented from electronic medical record notes and the Social Security Death Index [18]. Total RNA was previously submitted to the University of Michigan DNA sequencing core for library preparation and sequencing using the Illumina TruSeq Stranded Total RNA library prep kit. This data were previously summarized [19]. Here, to study viral content in greater detail, we leveraged the HPViewer pipeline using default settings to characterize HPV read counts in the cohort [20]. A previously defined threshold of > 5 reads was required to call a sample HPV positive.

### 2.2. Immunohistochemistry

Immunohistochemical staining was performed on the DAKO Autostainer (Agilent, Carpinteria, CA, USA) using Envision+ and diaminobenzadine (DAB) as the chromogen. De-paraffinized sections were labeled with the mouse p16 Ab-1 (DCS-50.1/47) (Neomarker, MS-218-P) and a mouse Ab (Thermofisher, Wyman Street, Waltham, MA, catalog #31430) was used as secondary antibody. Microwave epitope retrieval as specified was used prior to staining for all antibodies. Appropriate negative (no primary antibody) and positive controls (as listed) were stained in parallel with each set of slides studied. p16 immunostained slides were analyzed as previously described by our team [21].

### 2.3. HPV16 Capture-Based Targeted DNA Sequencing and Analysis

Formalin-fixed paraffin-embedded (FFPE) blocks were obtained for the single patient with HPV16+ MEC (as described below) and five additional patients without HPV reads by HPViewer, selected at random, for confirmatory NGS-based tumor analysis. Regions with >60% tumor content, as identified by our head and neck pathologist (J.B.M.), were identified for DNA isolation with the Qiagen Allprep DNA/RNA FFPE kit (Qiagen, Hilden, Germany). Using the DNA Thruplex kit for library preparation (Takara Biosciences), targeted capture sequencing on DNA that passed our quality control standards was performed by the University of Michigan Advanced Genomics Core as previously described [22]. We employed a custom-designed probe panel from Nextera that included high density probes covering the HPV16 genome as well as probes for targeting several common cancer-related genes [23]. Following library preparation and capture, the samples were sequenced on an Illumina NOVASeq6000 using a 300-cycle run and FastQ files were archived. HPV integrations were called by SearcHPV [24], an HPV integration caller that we recently developed for the detection of HPV-human integration loci from targeted capture DNA sequencing data. Downstream analysis was performed with R 3.6.1 and Python.

### 2.4. RNA-seq Data Analysis

Quality of the sequencing reads was evaluated using FastQC v.0.11.5. The quality reports did not reveal any adapter contamination; therefore, it was not considered necessary to perform quality trimming. The reads were mapped to the hg19 reference genome following a two-step alignment workflow of STAR v2.5.3a. Next, samtools v1.2 was used to extract uniquely mapped reads and Cufflinks v2.2.1 was used to generate the FPKM data. The *--max-bundle-frags* parameter of cufflinks was adjusted from its default value of 1,000,000 to 100,000,000 to allow us to compute FPKM at loci with high depth of coverage. Additionally, we applied SurVirus [25] and SearcHPV [24] to identify potential HPV-host fusions on the one HPV+ MEC sample (MEC1).

### 2.5. HPV Oncogene Expression Analysis

The first step in viral oncogene expression analysis was to build a reference genome of human and viral sequences. For this purpose, we used a modified version of the HPV16 genome, and its corresponding annotation file as described in [26]. The human genome sequence was obtained from the 1000 Genomes Project (Phase II). RSEM v1.3.3 was then used to build reference files of these human and modified HPV sequences. The same pipeline was also used to estimate gene expression levels from the HPV positive MEC (MEC1) RNA sequenced sample.

## 3. Results

We recently performed comprehensive transcriptome sequencing on 48 FFPE MEC tumors with a majority of tumors arising in the parotid. To now characterize the HPV content of tumors within this cohort, we analyzed the data using the HPViewer algorithm [20]. This analysis nominated only one of the forty-eight tumors as potentially HPV positive, with high HPV16 read counts (Figure 1A, Appendix A). We then evaluated p16 protein expression, a marker known to correlate with HPV status in oropharyngeal HNSCC, and found that MEC1 showed diffuse positive cytoplasmic and nuclear staining of p16 by immunohistochemistry, while none of the five selected HPV-negative MEC samples stained positive (Figure 1B), suggesting that p16 may also function as a marker of HPV status in MEC.

Clinically, MEC1 was a locally recurrent MEC of the anterior ethmoid sinuses in a 51-year-old male who underwent anterior sub-cranial resection with pathology showing high-grade MEC without perineural invasion and with negative margins. The patient had initially presented three years prior with a high grade MEC of the anterior ethmoid sinuses treated with subtotal resection followed by adjuvant radiation. He is now alive with no evidence of recurrent disease over 19 years out from salvage surgery. Our previous RT-PCR molecular sub-typing analysis of his tumor demonstrated the presence of a *CRTC1*-*MAML2* fusion [18].

To further validate our HPV annotation of this cohort, we performed PCR and Sanger sequencing of genomic DNA from MEC1, which confirmed the presence of HPV16 DNA (Appendix A) in this tumor. Accordingly, 0/5 of selected cases without HPV reads in our RNAseq data were also confirmed to lack HPV16 DNA by this method (data not shown). We then performed targeted capture NGS with a custom high-density HPV16 capture panel on the tumor DNA, which demonstrated high HPV16 read counts from MEC1, but not MEC23, which had no RNA-seq support for any HPV and served as a negative control (Figure 2). Analysis of host control genes in both of the targeted libraries confirmed that both were successfully sequenced to >500X depth (Appendix A).

To test for sites of HPV integration in MEC1, we used our recently described SearcHPV pipeline [24] to perform HPV-host integration analysis and identified 22 insertion sites in the host genome from targeted capture sequencing data (Figure 3A). Breakpoint sequence analysis of the integration sites indicated that most (21/22) HPV-host junctions in MEC1 have some degree of microhomology (Figure 3B). Further gene level analysis showed that 13 HPV integrations occurred in known genes, with an in-line insertion into the *TMEM163*, *HIP1,* and *SIRT1* genes and reverse orientation insertions in the remaining ten integrations. (Figure 3C, Appendix A). Seven of these genes were expressed at a lower level than the median of all MECs analyzed; five genes were expressed higher than the median; and one gene, RP11-354K1.1, was not expressed in any of the MECs (Figure 3D). Finally, expressed HPV-host integration transcripts were not identified from RNA-seq for MEC1 by two different callers, SearcHPV [25] and SurVirus [26].

## 4. Discussion

Our primary objective in the present study was to utilize our advanced bioinformatics pipelines to evaluate for the presence and physical state of transcriptionally active HPV in MECs of various major and minor salivary gland subsites. We found an exceedingly low prevalence of HPV (1/48 tumors, 2.1%) in our MEC cohort, with the single positive case of recurrent MEC of the anterior ethmoid sinus harboring transcriptionally active HPV16+ DNA with multiple complex integration events into various cancer-related genes. Concurrently, this tumor showed upregulation of p16 by IHC and altered expression of host genes affected by viral integration events. Our data raises several important points for the discussion of MEC tumor biology and the development of clinically useful, predictive, and prognostic biomarkers for this disease.

The etiologic role and prognostic implications of HPV in head and neck malignancies besides squamous cell carcinoma of the oropharynx remains a contentious and much-debated topic in our field [27]. Validation of HPV as a causative driver of other head and neck malignancies would have vast and exciting implications for treatment selection and prognostication. The data on HPV in major and minor salivary gland malignancies is inconclusive and limited by inconsistent HPV detection methods, small patient cohorts, and heterogeneous tumor histologies and subsites [15,16,17]. For example, a 2009 study by Vageli et al. utilized HPV L1 consensus PCR and RT-PCR to analyze HPV status in nine parotid gland tumors, including pleomorphic adenomas, Warthin’s tumor, and acinic cell carcinoma [28]. The authors reported the presence of HPV16 or HPV18 DNA in seven of nine (77.8%) tumors and posited that high-risk HPV may be an etiologic agent in various salivary gland neoplasms. This preliminary data subsequently inspired later studies by Brunner et al., Isaveya et al., and Bishop et al. on the potential role of HPV in MEC [15,16,17].

Our findings of rare HPV positivity in MEC are consistent with Bishop et al. [17]. The authors of that study analyzed 92 MECs of various subsites with the objective of determining the prevalence of transcriptionally active HPV and its co-occurrence with *CRTC1-MAML2* translocation. They utilized HPV E6/E7 RNA ISH to assay HPV status but failed to identify transcriptionally active virus in any of their samples, independent of *CRTC1-MAML2* fusion status. Overexpression of p16 was not evaluated in their cohort. In a much smaller sample of six minor salivary gland MECs, Brunner et al. used HPV16/18 DNA ISH and p16 IHC to show the presence of HPV16/18 DNA in two (33.3%) MECs of the oral cavity [15]. Interestingly, strong and diffuse nuclear and cytoplasmic p16 staining on IHC was seen in both of these tumors but also in three additional MECs negative for HPV16/18 DNA by ISH. Due to the small number of tumors analyzed, the 33.3% HPV positivity rate may represent sampling bias rather than a true prevalence of transcriptionally active virus in MECs.

Our data conflicts most notably with that of Isayeva et al. in which the authors found a strikingly high prevalence of HPV16/18 DNA in 49/98 (50%) of their MEC cohort [16]. The authors used several complementary methods of HPV detection, including nested RT-PCR, HPV16/18 E6/E7 immunofluorescence, and HPV L1 consensus PCR to support their conclusion that high-risk HPV is convincingly implicated in the pathogenesis of MEC. However, their quoted HPV prevalence rate was based on nested RT-PCR only and they reported moderate discordance between HPV16/18 E6/E7 detection via immunofluorescence and their other detection methods. Further, the authors found no statistical correlation between HPV16/18 DNA detection via nested RT-PCR and p16 overexpression, tumor subsite, and tumor grade. Ultimately, the validity of their conclusions remains uncertain due to failure to replicate this prevalence rate in multiple independent studies and lack of correlative p16 overexpression indicating biologically relevant HPV infection in MEC [15,17,29]. An advantage of our study over previous ones is the use of contemporary, high-throughput, sophisticated, and highly sensitive bioinformatics pipelines for the detection of HPV DNA and viral transcription [24,25,26].

Herein, we conclude that transcriptionally active HPV is a rare occurrence in MEC, independent of tumor subsite. Previous studies have failed to show a higher prevalence of HPV in MECs of anatomic subsites within lymphoid tissue of Waldeyer’s ring. Further, there seems to be no predilection for HPV positivity in mucosal subsites or minor versus major salivary glands. Our single HPV-positive tumor was a recurrent MEC of the anterior ethmoid sinuses. It is interesting to speculate that, although an infrequent event overall, MECs of sinonasal subsites may be more prone to harbor HPV as a molecular driver. However, we cannot definitively reach that conclusion with our data.

Histologically, MECs are characterized by a variable, triphasic pattern of mucinous, intermediate, and epidermoid cells with histologic grade dependent on degree of nuclear atypia, mitoses, necrosis, perineural invasion, and cystic components [30]. Previous authors have not found any recurring histologic features characteristic of HPV-positive MECs that may differentiate these tumors from their non-HPV associated counterparts [16]. Further, transcriptionally active HPV does not seem to localize to mucinous, intermediate, or epidermoid components preferentially. Similarly, our single HPV-positive MEC did not harbor any distinguishing histologic features that may be of routine clinical utility (Figure 1). Thus, we conclude that the presence of transcriptionally active HPV does not confer a distinguishable histologic pattern of MECs nor reliably impact tumor grade.

When MECs harbor transcriptionally active HPV however, the virus may significantly alter host gene expression by complex viral integration mechanisms. We showed that HPV integrated into 13 host genes, including *PIK3AP1, SIRT1*, *ARAP2*, *TMEM161B-AS1*, and *EPS15L1* as well as 9 non-genic regions. Interestingly, the PI3K pathway has been implicated in MECs, and alterations were identified in 52% of high-grade cases in a targeted sequencing analysis of one cohort, which was consistent with RNAseq analysis of 8 high-grade tumors showing a re-programming of PI3K signaling effectors [31]. Previous mechanistic data has shown that *PIK3AP1* expression drives AKT phosphorylation in gastric and thyroid cancer models, suggesting an oncogenic role of this gene in other cancers [32,33]. Notably, however, transcriptional analysis of MEC1 suggested that the integration events showed no significant patterns on the impact of the expression of these genes. Thus, we expect that the causal mechanism by which HPV contributes to pathogenesis in the HPV+ tumor identified in our cohort is through elevated E6 and E7 oncogene expression leading to p16 overexpression.

Despite this notion, viral integration into *SIRT1* is also of potential pathogenetic interest to MEC. *SIRT1* has a complex and multi-faceted role in cancer that includes the regulation of *TP53* as well as responses to DNA damage, metabolism stress, and inflammation [34]. *SIRT1* is a NAD^+^-dependent Class III histone deacetylase that has been shown to antagonize cellular senescence [35] and is also an established negative regulator of *CRTC2*, and possibly *CRTC1* as well [36], suggesting that this integration event could enhance *CRTC1*-*MAML2* expression in this tumor. Likewise, in the context of mutant p53, *SIRT1* acts as a tumor suppressor [37]. Given the established role of HPV16_E6 in repressing p53 activity, it is possible that *SIRT1* could act as a tumor suppressor in the context of HPV16+ MEC1, consistent with the relatively low expression of *SIRT1* observed in this sample. To our knowledge, *ARAP2*, *TMEM161B-AS1*, and *EPS15L1* have not previously been indicated as playing a role in MEC pathogenesis. As future genetic analyses of MEC tumors are published, it will be interesting to see if any of these genes disrupted by HPV integration are also altered by other genetic mechanisms, as the observation of multiple mechanisms of genetic disruption would support a critical role for the genes.

In conclusion, using our integrative sequencing analysis, we demonstrate that transcriptionally active HPV is a rare occurrence in MEC. However, when present, HPV can have a substantial role in altering the host genome, including through the direct integration and disruption of host genes. Our data suggest that alternative drivers other than HPV are much more frequently responsible for the pathogenesis of MEC, including *CRTC1/3-MAML2* fusion negative cases. From a long-term perspective, while the prevalence of HPV-related cancers is increasing around the world, our data suggests that HPV-related MECs are likely to account for only a minor subset of this disease. Our data is consistent with recently published series in concluding that no particular MEC subsite is more prone to harbor transcriptionally active HPV. Indeed, while our data resolve fundamental questions about the role of HPV in this disease, it also supports a need for future research to help identify additional genetic drivers of MEC. Due to the rarity of transcriptionally active HPV in MEC, we cannot advocate for routine p16 or HPV DNA ISH testing in clinical practice.

## Figures and Tables

**Figure 1 viruses-14-02353-f001:**
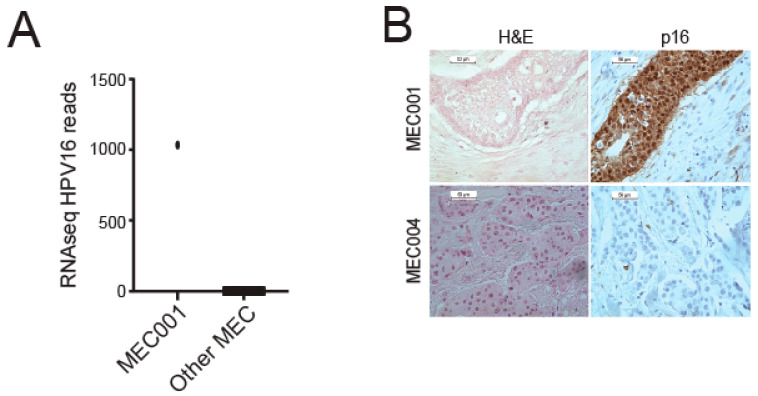
Analysis of HPV type distribution in our MEC cohort. (**A**) Number of RNA-seq HPV16 reads for MEC1 and other MECs. (**B**) p16 immunohistochemistry performed on sections from the HPV16+ tumor (MEC1) and a representative HPV negative tumor (MEC4).

**Figure 2 viruses-14-02353-f002:**
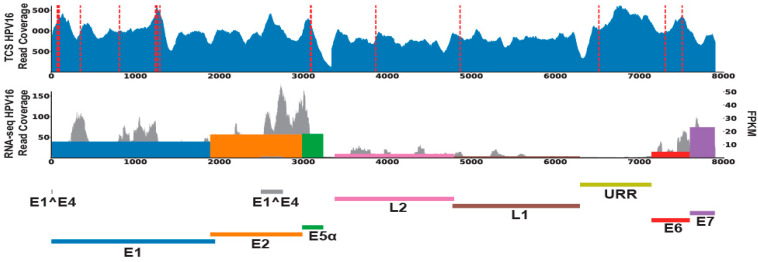
HPV16 DNA and RNA content in MEC1. The read coverages of HPV for MEC1 were filled in blue (first lane) for targeted capture sequencing data; grey for RNA-seq data (second lane). Red dashed lines denoted the HPV-host integrations called from targeted capture sequencing data. HPV genes expression levels (FPKM) were marked as colored bars in the second lane.

**Figure 3 viruses-14-02353-f003:**
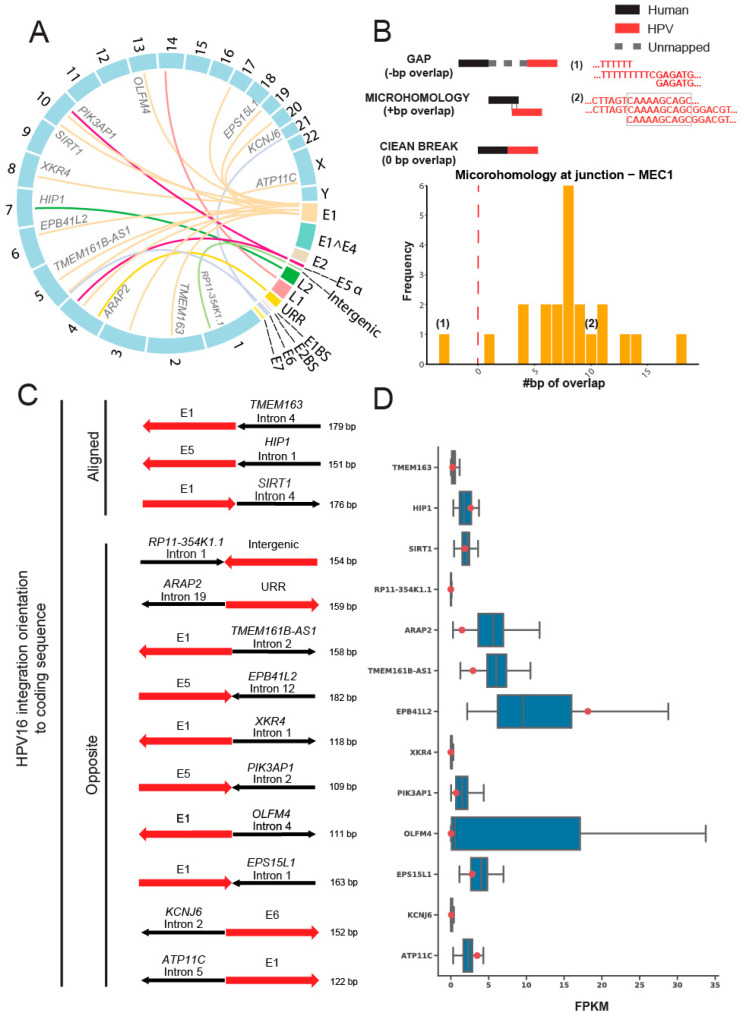
HPV16 integration site analysis in the host genome of MEC1. (**A**) Link plot of the HPV-host integrations in MEC1. Lines were colored by HPV genes. Host genes that integrations fell into were marked. (**B**) Microhomology at HPV-host junctions in MEC1. The microhomology was defined as the overlapped base pairs between human and HPV segments at the junctions. Overlapped bases referred to positive scores of microhomology, e.g., example (1); gaps referred to negative scores, e.g., example (2), and clean ends referred to zeros. (**C**) HPV integration orientation to coding sequence. The human segments were colored in black and HPV segments were colored in red. Arrows of human segments indicated the human gene orientation: right, positive strands; left, negative strands. Arrows of HPV segments pointed the HPV gene orientation in contrast to the corresponding human gene. (**D**) RNA expression levels of genes that HPV integrations fell into. Blue box plots denoted the FPKM of all 48 MECs for the 13 genes that HPV integrations fell into. Red points showed the FPKM for MEC1. Note that FPKM at RP11-354K1.1 were zero for all MECs.

## Data Availability

All data is available from the senior author upon reasonable request.

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
