# Peer review of "Analysis of Human Papilloma Virus Content and Integration in Mucoepidermoid Carcinoma"

_viruses, 2022, doi:10.3390/v14112353_

Round 1

Reviewer 1 Report

This is a well constructed study , although it demonstrates only 1 of 48 cases to be related to HPV, it is very worthy of publication and adds to our knowledge of the HPV role in tumour genesis.

Author Response

Thank you for your positive comments on our manuscript. 

Reviewer 2 Report

The manuscript “Analysis of Human Papilloma Virus Content and Integration in Mucoepidermoid Carcinoma” presents evidence which argues for infrequent involvement of human papillomavirus (HPV) in a subset of head and neck squamous cell carcinomas, mucoepidermoid carcinomas (MECs). The authors employ bioinformatics analyses on a cohort of 48 MECs which had been previously subjected to transcriptome analysis. Using this methodology, they detect HPV16 reads in a single tumor from their cohort.

To validate presence of HPV sequences in the one sample (and absence in a subset of the others) they performed PCR, followed by Sanger sequencing, as well as p16 staining for which positivity is often to correlate with HPV positivity. The authors further characterize integration sites for HPV within 13 genes and 9 intergenic regions. While the authors discuss the potential involvement of those integrations in the pathogenesis of MEC they do not discuss other potential evidence for the involvement of HPV in the pathogenesis of a tumor (eg. absence of p53 mutations in HPV-positive tumor vs frequent mutations in the HPV-negative ones, APOBEC signatures etc).

Nevertheless, their work provides useful insight regarding a subset of HNSCC where previously the involvement of HPV positivity was suggested by some studies to be far more substantial. Their findings, despite the small cohort, and when considered in light of other relevant literature would suggest that while the positivity of HPV is likely an infrequent event in MECs, it may represent an involvement contributing to carcinogenesis as in other HNSCC. These findings are thoroughly discussed in the conclusions within the broader context of similar studies. This reviewer would suggest that the authors either provide further evidence or at least discussion which may point to or away from the causative nature of HPV in this tumor.

Author Response

We thank the reviewer for the positive comments about our manuscript. We have added the following sentence in response to the request for additional discussion. 

"Thus, we expect that the causal mechanism by which HPV contributes to pathogenesis in the HPV+ tumor identified in our cohort is through elevated E6 and E7 oncogene expression leading to p16 overexpression. "